# Estimating Solar Irradiance Using Sky Imagers

Soumyabrata Dev[1,2], Florian M. Savoy[3], Yee Hui Lee[4], and Stefan Winkler[3,5]

[1]ADAPT SFI Research Centre, Dublin, Ireland
[2]School of Computer Science, University College Dublin, Ireland
[3]Advanced Digital Sciences Center (ADSC), University of Illinois at Urbana-Champaign, Singapore 138632
[4]School of Electrical and Electronic Engineering, Nanyang Technological University (NTU), Singapore 639798
[5]School of Computing, National University of Singapore

*Correspondence to:* Stefan Winkler (winkler@comp.nus.edu.sg)

**Abstract.** Ground-based whole sky cameras are now-a-days extensively used for localized monitoring of the clouds. They capture hemispherical images of the sky at regular intervals using a fisheye lens. In this paper, we propose a framework for estimating the solar irradiance using pictures taken by those imagers. Unlike pyranometers, these sky images contain information about the cloud coverage and can be used to derive cloud movement. An accurate estimation of the solar irradiance using solely those images is thus a first step towards short-term solar energy generation forecasting, as cloud movement can also be derived from them. We derive and validate our model using pyranometers co-located with our whole sky imagers. We achieve a better performance in estimating the solar irradiance, as compared to other related methods using ground-based observations. Our method shows a significant improvement in estimating strong short-term variations, as compared to methods Hargreaves and Samani (Hargreaves and Samani, 1985), Bristow and Campbell (Bristow and Campbell, 1984), Donatelli and Campbell (Donatelli and Campbell, 1998) and Hunt *et al.* (Hunt et al., 1998).

## 1 Introduction

Clouds have a significant impact on solar energy generation. They intermittently block the sun and significantly reduce the solar irradiance reaching solar panels. A short-term forecast of the solar irradiance is needed for the grid operators to mitigate the effects of a power generation ramp-down. With rapid developments in photogrammetric techniques, ground-based sky cameras are now widely used (Dev et al., 2016e). These cameras, known as Whole Sky Imagers (WSIs) are upward looking devices that captures the sky scene at regular intervals of time. The images captured by WSIs are subsequently used for automatic cloud coverage computation, cloud tracking and cloud base height estimation. In our research group, we use these imagers to study the effect of clouds in satellite communication links (Dev et al., 2018b; Yuan et al., 2016; Dev et al., 2016a).

Localized and short-term forecasting of cloud movements is an on-going research topic (Shakya et al., 2017; Jiang et al., 2017; Feng et al., 2018). Optical flow techniques can be used to generate forecasted images using two anterior frames (Dev et al., 2016d). Similar cloud motion vectors are exploited in satellite images for solar power prediction (Jang et al., 2016). Our proposed method in estimating solar irradiance is thus a first step towards solar irradiance forecasting, as the input data used to estimate the irradiance is the same as the one used to forecast the sky condition.

The accurate estimation of solar energy is a challenging task, as clouds greatly impact the total irradiance received on the earth's surface. In the event of clouds covering the sun for a short time, there is a sharp decline of the produced solar energy. Therefore, it is of utmost importance to model the incoming solar radiation accurately. In this paper, we answer this fundamental question: can the rapid fluctuations of the solar irradiance be captured? We perform this by using ground-based 5 sky cameras to estimate the solar irradiance.

The analysis of clouds and several other atmospheric phenomenon is traditionally done using satellite images. However, these satellite images have low temporal and spatial resolutions. The most widely used satellite data is from Moderate-resolution Imaging Spectroradiometer (MODIS) (Pagano and Durham, 1993), which is on board the Terra and Aqua satellites. They provide a large-scale view of the cloud dynamics and various atmospheric phenomenons. The data from this satellite on-board 10 instruments are usually available only twice in a day. This is useful for a macro-analysis of cloud formation at a particular location on the earth's surface. One of the illustrative examples of such satellite data is the HelioClim-1 database from Global Earth Observation System of Systems (GEOSS) (Lautenbacher, 2006). It provides hourly and daily average of surface solar radiation received at ground level (Lefèvre et al., 2014). Ouarda et al. in (Ouarda et al., 2016) assessed the solar irradiance from six thermal channels obtained from Spinning Enhanced Visible and Infrared Imager (SEVIRI) instrument. However these 15 information are temporal and spatial averages. Solar energy applications requires knowledge of the solar irradiance at specific locations and at every time throughout the day. Therefore, images obtained from satellite are not conducive for analysis, especially in geographical small countries like Singapore where the cloud formation is highly localized.

## 1.1 Related Work

Several existing works analyze ground-based images with different meteorological observations. Most of them correlate the 20 cloud coverage obtained from the sky images with the human observations from meteorological centers. Silva and Souza-Echer validated cloud coverage measurements obtained from ground-based automatic imager and human observations for two meteorological stations in Brazil (Silva and Souza-Echer, 2016). Huo and Lu also performed such field experiments for three sites in China (Huo and Lu, 2012). The computation of such cloud coverage percentage is important in solar energy generation. It can hugely impact the amount of solar radiation falling at a particular place. The correct estimation of solar irradiance, is 25 particularly important in tropical countries like Singapore, where the amount of received solar irradiance is high. Rizwan et al. in (Rizwan et al., 2012) have demonstrated that tropical countries are conducive for installing large central power stations powered by solar energy, because of the large amount of incident sunlight throughout the year. Several attempts have been done to estimate the solar radiation from general meteorological measurements via temperature, humidity and precipitation (Hargreaves and Samani, 1985; Donatelli and Campbell, 1998; Bristow and Campbell, 1984; Hunt et al., 1998). These existing 30 models aim to provide global solar radiation using different sensors. Alsadi and Nassar in (Alsadi and Nassar, 2017) has demonstrated such estimation models from the perspective of a photovoltaic solar field. They have effectively demonstrated that the succeeding rows in a photovoltaic solar field receive less solar radiation than that of first row. They also provided an analytical solution by including the design parameters in the estimation model. In addition to solar irradiance estimate, there have been several efforts in forecasting the solar irradiance with a lead time of few minutes. Baharin *et al.* proposed a machine-

learning forecast model for PV power output, using Malaysia as the case study (Baharin et al., 2016). Similarly Chu *et al.* used a reforecasting method to improve the PV power output forecasts with a lead time of 5, 10, and 15 minutes (Chu et al., 2015). Satellite images have also been used in the realm of solar analytics. Mueller *et al.* proposed a clear sky model that is based on radiative transfer models obtained from Meteosat's atmospheric parameters (Mueller et al., 2004). However, satellite data have lower temporal and spatial resolutions. Recently, with the development of low-cost photogrammetric techniques, sky camera are being deployed for such purposes. These sky cameras have a higher temporal and spatial resolutions, and provide a more localized information about the atmospheric events. Alonso-Montesinos and Batlles used sky cameras to quantify the total solar radiation (Alonso-Montesinos and Batlles, 2015). Yang and Chen studied these solar irradiance variability using entropy and covariance (Yang and Chen, 2015). Dev et al. in (Dev et al., 2018a) used triple exponential smoothing for analyzing the seasonality of the solar irradiance. However, these approaches could not model the sharp short-term variations of solar radiation.

## 1.2 Outline of our work

In this paper, we use images obtained from WSIs to accurately model the fluctuations of the solar radiation. There are several advantages of using a WSI to estimate solar irradiance, instead of using a pyranometer. Common weather stations generally uses a solar sensor that measures the total solar irradiance. It is a point-source device providing information for a particular location. It does not provide information on cloud macrophysical properties, and its evolution over time. On the other hand, the wide-angle view of ground-based sky camera provide us extensive information of the sky. It allows for the tracking of cloud mass over successive image frames, and also predict its future location. In this paper, we attempt to solve the fundamental problem of modeling solar irradiance from sky images. This will also help in solar energy forecasting, which is useful in photovoltaic (PV) systems (Lorenz et al., 2009).

The main contributions of this paper are as follows:

- A robust framework to accurately estimate and track the rapid fluctuations of solar irradiance;

- A proposal of estimating solar irradiance using ground-based sky camera images;

- An extensive benchmarking of our proposed method with other solar irradiance estimation models.

The rest of the paper is organized as follows. Section 2 describes our experimental setup that captures the sky/cloud images and collects other meteorological sensor data. Our framework for estimating solar irradiance is detailed in Section 3. Section 4 discusses the evaluation of our approach, and its benchmarking with other existing solar estimation models. We discuss the possible applications of our approach in Section 5. We also point out a few limitations of our approach, and ways to address them. Section 6 concludes the paper.

## 2 Data Collection

Our experimental setup consists of weather stations and ground-based WSIs. These devices are collocated at the rooftop of our university building (1.34°N, 103.68°E). These devices continuously capture the various meteorological data, and archive them for subsequent analysis.

## 2.1 Whole Sky Imager (WSI)

Commercial WSIs are available in the market. However, those imagers have high cost, low image resolution, and less flexibility in operation. In our research group, we have designed our custom-built, low-cost, and high-resolution sky imagers. These imagers are called WAHRSIS, that stands for Wide Angle High Resolution Sky Imaging System (Dev et al., 2014). A WAHRSIS imager essentially consists of a high-resolution digital single-lens reflex (DSLR) camera with a fish-eye lens and an on-board micro-computer. The DSLR camera has a digital imaging sensor, instead of the traditional photographic film. The entire device is sealed inside a box with a transparent dome for the camera. Over the years, we have built several versions of WAHRSIS (Dev et al., 2014, 2015). They are now deployed at several rooftops of our university campus, capturing images of the sky at intervals of 2 minutes.

Our ground-based camera WAHRSIS is calibrated with respect to – white balancing, geometric calibration and vignetting correction. The imaging system in WAHRSIS is modified so that it captures the near-infrared region of the spectrum. Hence, the red channel of the captured image is more prone to saturation. It renders the captured image reddish in nature. Therefore, we employ custom white balancing in the camera, such that it compensates the alteration owing to the near-infrared capture. Figure 1 depicts the captured images obtained from automatic and custom white balancing.

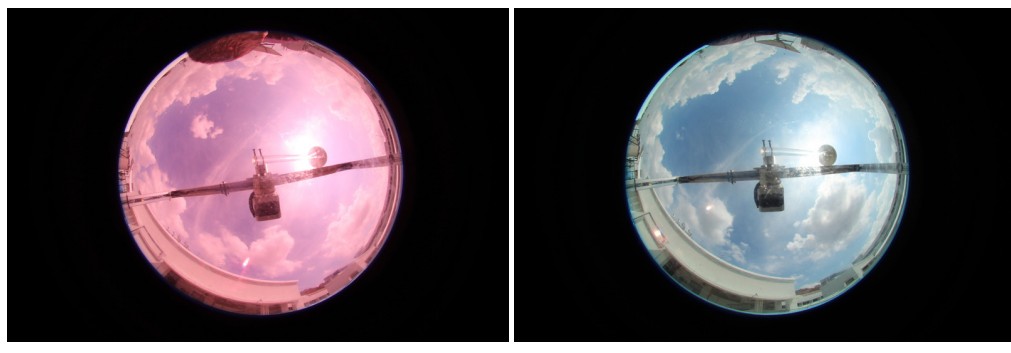

(a) With auto white-balancing        (b) With custom white-balancing

**Figure 1.** We use custom white-balancing for correcting the white balance.

We use the popular toolbox by Scaramuzza *et al.* (Scaramuzza et al., 2006) for the geometric calibration of WAHRSIS. This process involves the computation of the intrinsic parameters of the camera. We use a black-and-white regular checkerboard pattern, and position it at various positions around the sky camera. Figure 2 illustrates a few sample positions of the checker-

board. Using user interaction to identify the corner points and the known 3D co-ordinates, we estimate the intrinsic parameters of the camera.

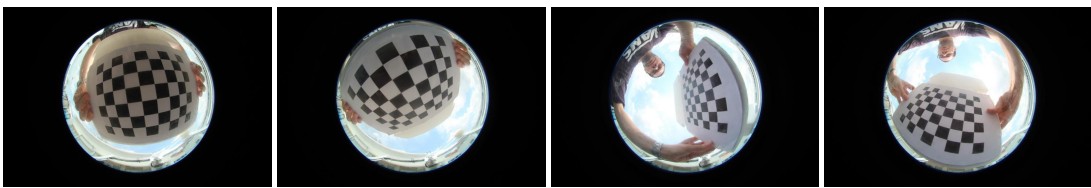

**Figure 2.** We position the checkerboard at various locations for the geometric calibration.

Finally, we also employ vignetting correction to the images captured by our sky camera. Owing to the fish-eye nature of the lens, the area around the centre of the lens is brighter, as compared to the sides. We use an integrating sphere to correct this

5    variation of illumination. Figure 3 depicts an image captured inside an integrating sphere that provides an uniform illumination distribution in all directions. We use this reference image to correct the illumination of all captured sky/cloud images by our sky camera.

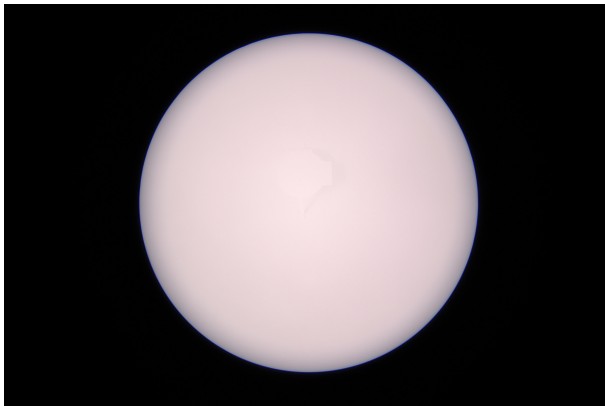

**Figure 3.** We captured a reference image inside the uniformly-illuminated integrating sphere.

## 2.2    Weather Station

In addition to the sky imagers, we have also installed collocated weather stations. We use *Davis Instruments 7440 Weather*

10    *Vantage Pro* for our recordings. It measures rainfall, total solar radiation, temperature and pressure at intervals of 1 minute. The resolution of the tipping-bucket rain gauge is 0.2 mm/tip.

It also includes a solar pyranometer measuring the total solar irradiance flux density in Watt/m$^2$. This consists of both direct and diffused solar irradiance component. The solar sensor integrates the solar irradiance across all angles, and provide the net solar irradiance. On a clear day with no occluding clouds, the solar sensor ideally follows a typical cosine response. Figure 4

shows the theoretical response of the solar sensor in the pyranometer, for varying degrees of solar incident angle. The solar sensor reading is highest during noon when the incident angle of sun rays is at the minimum, whilst the reading is low during morning and evening hours.

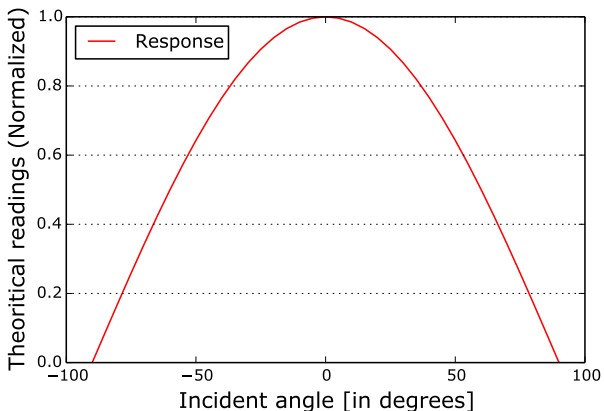

**Figure 4.** Response of the solar sensor with varying solar incident angle.

The solar radiation on a clear sky can be modeled using the solar zenith angle, and earth's eccentricity. Several clear sky
models have been developed for various regions. The best clear-sky model for Singapore is provided by Yang *et al.* (Yang et al., 2012). We performed a comparison of various clear sky models in Singapore (Dev et al., 2017), and found that the Yang *et al.* provides a good estimate of the clear sky irradiance. The clear-sky Global Horizontal Irradiance (GHI) $G_c$ is modeled as:

$$G_c = 0.8277 E_0 I_{sc} (\cos \alpha)^{1.3644} e^{-0.0013 \times (90 - \alpha)}, \tag{1}$$

where $E_0$ is the eccentricity correction factor for earth, $I_{sc}$ is the solar irradiance constant (1366.1Watt/m$^2$), and $\alpha$ is the
solar zenith angle (measured in degrees). The factor $E_0$ is calculated as:

$$E_0 = 1.00011 + 0.034221 \cos(\Gamma) + 0.001280 \sin(\Gamma) + 0.000719 \cos(2\Gamma) + 0.000077 \sin(2\Gamma),$$

where $\Gamma = 2\pi(d_n - 1)/365$ is the day angle (measured in radians) and $d_n$ is the day number of the year.

As an illustration, we show the clear-sky radiation for the 1st of September 2016 in Fig. 5. The actual solar radiance measured by our weather station is also plotted. We also show the deviation of the measured solar radiation from the clear-sky model.
We observe that there are extremely rapid fluctuations in the measured readings. In our previous work (Dev et al., 2016b), we observed that these rapid fluctuations caused by the incoming clouds that obstruct the sun from direct view. Such information about the cloud profile and its formation cannot be obtained from a point-source solar recording. Therefore, we aim to model these rapid fluctuations in the measured solar radiation from wide-angle images captured by our sky cameras.

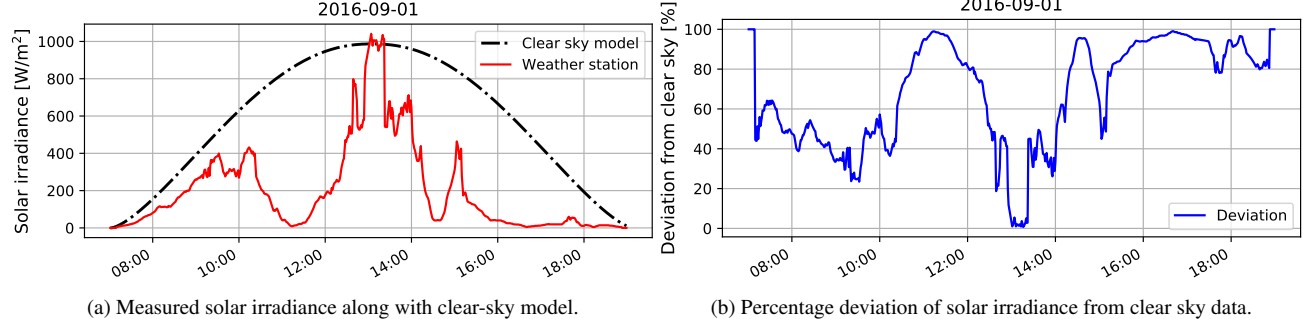

(a) Measured solar irradiance along with clear-sky model.    (b) Percentage deviation of solar irradiance from clear sky data.

**Figure 5.** Solar irradiance measurements on the 1st of September 2016. Note the rapid fluctuations of high magnitude in the measured solar irradiance recording.

## 3 Modeling Solar Irradiance

This section details our model for computing the solar irradiance from images captured by a whole sky imager. We sample pixels using a cosine weighted hemispheric sampling to simulate the behavior of a pyranometer based on the fisheye camera lens. We then compute the relative luminance using the image capturing parameters, after gamma correction. We finally derive
a empirical fitting function to scale the computed luminance to match measured irradiance values.

### 3.1 Cosine weighted hemispheric sampling

The behavior of our fisheye lens with focal length $f$ is modeled by the equisolid equation $r = 2f \sin(\theta/2)$, relating the distance $(r)$ of any pixel from the center of the image to its incident light ray elevation angle $(\theta)$. This allows to project a captured image on a unit hemisphere, as shown in Fig. 6a.
The solar irradiance is composed of a direct component relating the sun light reaching the earth without interference, as well as diffuse and reflected components. Given the high resolution of our images, we consider randomly sampled pixel locations on the hemisphere as input to the luminance computation. We follow a cosine weighted hemispheric distribution function, the center of which is at the location of the sun. This is because clouds in the circumsolar region have the highest impact on the total solar irradiance received on the earth's surface (Dev et al., 2016b). We provide more emphasis to the clouds around the
sun, as compared to those near the horizon. In our previous work (Dev et al., 2016c), we used a cloud mask around the sun to estimate the solar irradiance. However, such methods needs an additional task of optimizing the size of the cropped image for best results. Therefore, we adopted this strategy of cosine weighted hemispheric sampling.

  The first step is to compute the sampled locations from the top of the unit hemispheric dome. Each of the locations are computed as follows, using two random floating points $R_1$ and $R_2$ as input, where $(0 \leq R_1, R_2 \leq 1)$:

$\phi = 2\pi R_1, \ \theta = \arccos(\sqrt{R_2})$

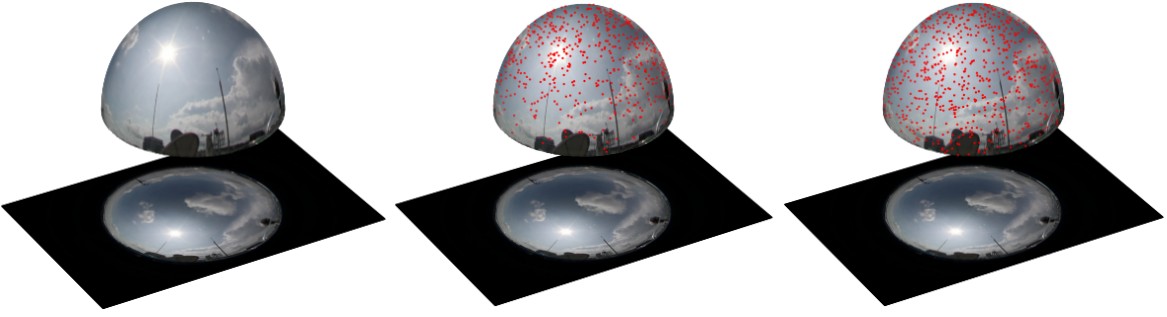

(a) Projection on a hemisphere of the original image

(b) Cosine hemispheric sampling of the hemisphere with origin on the top

(c) Applying a rotation matrix to center at the sun location

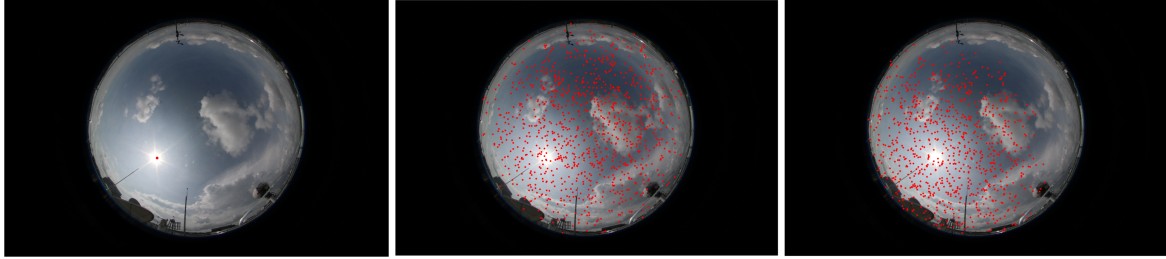

(d) Original image with detected sun location in red

(e) Projection on the image of the sampled points

(f) Projection on the image of the rotated sampled points

**Figure 6.** Cosine weighted hemispheric sampling process used to select the pixels used for solar irradiance estimation

$$
\begin{bmatrix} x \\ y \\ z \end{bmatrix} = \begin{bmatrix} \sin(\theta) \cdot \cos(\phi) \\ \sin(\theta) \cdot \sin(\phi) \\ \cos(\theta) \end{bmatrix} \tag{2}
$$

This is represented in Fig. 6b.

The second step is to detect the location of the sun using a thresholding method. This is needed to align the center of the previously computed distribution (i.e. top of the hemispheric dome) to the actual sun location in the unit sphere. We choose a threshold of $240$ in the red channel $R$ of the $RGB$ captured image, and compute the centroid of the largest area above the threshold (Savoy et al., 2016). We then compute the rotation matrix transforming the z-axis unit vector to the unit vector pointing towards the sky. We apply this rotation to all the sampled points, resulting in Fig. 6c.

This means that the amount of sampled points in a region of the hemisphere is proportional to the cosine of the angle between the sun direction and the direction to that region. We experimentally concluded that this achieves a good balance between all irradiance components. We thus consider the pixel values of $5000$ points sampled using this method as input for the irradiance estimation.

## 3.2 Relative luminance calculation

For each of the $i$ sampled pixels in the $RGB$ image, we compute its luminance value using the following formula. The formula is proposed in SMPTE Recommended Practice 177 (SMPTE, 1993). It is used to compute the luminance of an image from the $R$, $G$ and $B$ values of the $RGB$ image.

$$Y_i = 0.2126 \cdot R_i + 0.7152 \cdot G_i + 0.0722 \cdot B_i \qquad 5$$

The JPEG compression format encodes images after applying a gamma correction. This non-linearity mimics the behavior of the human eye. This needs to be reversed in order to compute the irradiance. We use a gamma correction factor of 2.2, which is most commonly used in imaging systems (Poynton, 2003). We thus apply the following formula, assuming pixel values normalized between 0 and 255:

$$Y_i' = 255(Y_i/255)^{2.2} \qquad 10$$

We then average the pixel values across all the $i$ sampled points in the image, and denote it by $\mathcal{N}$. This pixel value given by $\mathcal{N} = (1/n)\sum_{i=1}^{n} Y_i'$, denotes the average luminance value of the sampled points from the image.

However, each image of the sky camera is captured with varying camera parameters via ISO, F-number and shutter speed. These camera parameters can be read from the image metadata, and are useful to estimate the scene luminance. The amount of brightness of the sampled points $\mathcal{N}$, is proportional to the number of photons hitting the camera sensor. This relationship between scene luminance and pixel brightness is linear (Hiscocks and Eng, 2011), and can be modeled using the camera parameters as:

$$\mathcal{N} = K_c \left( \frac{e_t \cdot S}{f_s^2} \right) \mathcal{L}_s$$

where $\mathcal{N}$ is the pixel value, $K_c$ is a calibration constant, $e_t$ the exposure time in seconds, $f_s$ the aperture number, $S$ the ISO sensitivity and $\mathcal{L}_s$ the luminance of the scene.

We can thus compute the relative luminance $\mathcal{L}_r$ using the following:

$$\mathcal{L}_r = \mathcal{N} \left( \frac{f_s^2}{e_t \cdot S} \right)$$

## 3.3 Modeling irradiance from luminance values

Using our hemispheric sampling and relative luminance computation, we therefore have one relative luminance value $\mathcal{L}_r$ per image. We propose our model using this relative luminance value to estimate the solar radiation. The usual sunrise time in Singapore is between 6:40 am and 7:05 am, and sunset time is approximately between 6:50 pm and 7:10 pm. This information is obtained from (nea). Therefore, we consider images captured in the time interval of 7:00 am till 7:00 pm. We use our ground-based whole sky images captured during the time period from January 2016 till August 2016 to model the solar

radiation. The solar irradiance is computed as the flux of radiant energy per unit area normal to the direction of flow. The first step in estimating irradiance from the luminance is thus to cosine weight it according to its direction of flow. We weight our measurements according to the solar zenith angle $\alpha$. This is based on empirical evidences of our experiments on solar irradiance estimation. The modeled luminance $\mathcal{L}$ is expressed as:

$$\mathcal{L} = \mathcal{L}_r(\cos \alpha)$$

Let us assume that the actual solar radiation recorded by the weather station be $\mathcal{S}$. We check the nearest weather station measurement, for all the images captured by WAHRSIS between April 2016 till December 2016. Figure 7 shows the scatter plot between the image luminance and solar radiation. The majority of the data follows a linear relationship between the two. However, it deviates from linearity for higher values of luminance. This is mainly because of the fact that the mapping between scene luminance and obtained pixel value in the camera sensor becomes non-linear for large luminances. A more detailed discussion on this is provided in Section 5.

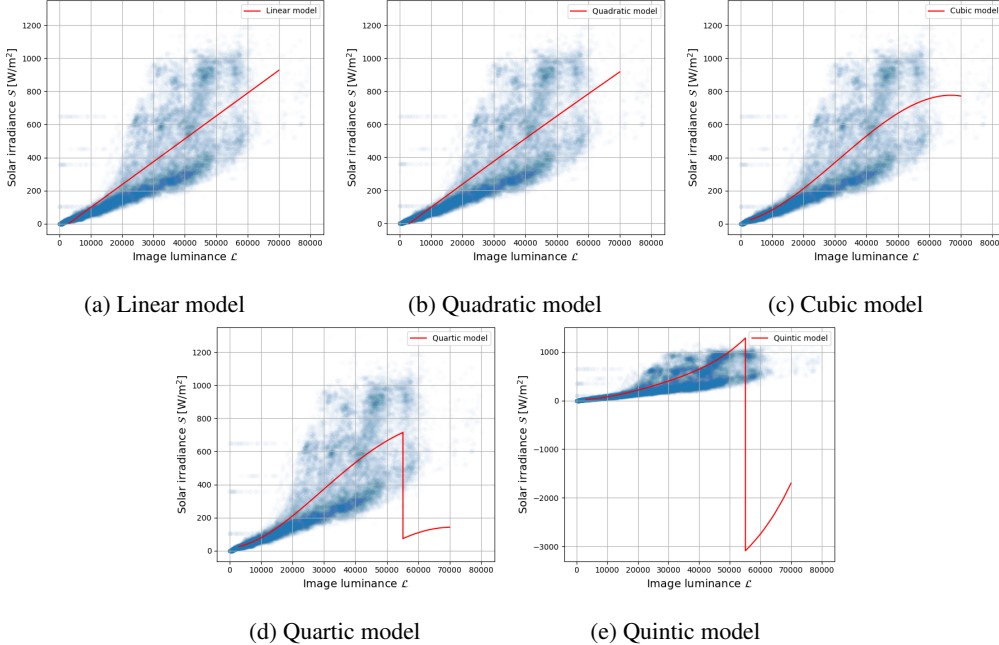

(a) Linear model      (b) Quadratic model      (c) Cubic model

(d) Quartic model      (e) Quintic model

**Figure 7.** Empirical fit between solar irradiance and image luminance computed with our proposed framework. We observe that it deviates from linearity at higher luminance values. Also, the higher order polynomials are ill-conditioned.

We attempt to fit a linear model and other higher-order polynomial regressors to model the relationship between image luminance from sky camera images and the measured solar radiation. Figure 7 shows the best fit line for several orders of polynomial function. In order to provide an objective evaluation of the different models, we also compute the RMSE value

between the actual- and regressed- values. Table 1 summarises the performance of the different order polynomials. We observe that lower order polynomials of degree 1 and 2 perform slightly inferior to those of higher order polynomials. We observe that the performance of cubic, quartic and quintic models are similar. However, from Fig. 7, we observe that the quartic and quintic models are highly ill-conditioned. Therefore, we choose the cubic model as our proposed model to model the measured solar radiation $\mathcal{S}$ from the image luminance $\mathcal{L}$. This is based on our assumption that the mapping from scene luminance to pixel values in the captured image is linear for lower luminance values, and it behaves in a non-linear fashion for higher luminance values. We use this selected model in all our subsequent discussions and evaluations.

| Proposed models | RMSE (W/m$^2$) |
|---|---|
| Linear model (degree 1) | 178.27 |
| Quadratic model (degree 2) | 178.26 |
| Cubic model (degree 3) | 176.57 |
| Quartic model (degree 4) | 176.52 |
| Quintic model (degree 5) | 176.49 |

**Table 1.** Performance evaluation of various polynomial order regressors. We measure the RMSE value for each of the models.

We model solar radiation as: $\mathcal{S} = a_3 \times \mathcal{L}^3 + a_2 \times \mathcal{L}^2 + a_1 \times \mathcal{L} + a_0$. The values of $a_3$, $a_2$, $a_1$ and $a_0$ are derived as $-4.25e-12$, $3.96e-07$, $0.00397$ and $7.954$ respectively for our data. Therefore, our proposed methodology in estimating solar irradiance from luminance is:

$$\mathcal{S} = (-4.25e-12)\mathcal{L}^3 + (3.96e-07)\mathcal{L}^2 + (0.00397) \times \mathcal{L} + 7.954. \tag{3}$$

This model is derived specifically for equatorial region like Singapore, and the regression constants are based on our WAHRSIS sky imaging system. However, these values need to be fine-tuned while applying our methodology for other regions and different imaging system. The source code of all the simulations in this paper is open-source, and the code repository is available online at https://github.com/Soumyabrata/estimate-solar-irradiance.

## 4 Performance Comparison and Validation

In this section, we evaluate the accuracy of our proposed approach. It is derived based on WAHRSIS images captured from January to August 2016. We also use these images to evaluate the accuracy of our proposed model. Furthermore, we benchmark our algorithm with other existing solar radiation estimation models.

### 4.1 Evaluation

One of the main advantages of our approach is that all rapid fluctuations of solar radiation can be accurately tracked from the image luminance. We illustrate this by providing the measured solar readings of 01-Sep-2016 in Fig. 8. The clear-sky

model follows a cosine response and is shown in black; whereas the measured solar recordings is shown in red. We normalize our computed luminance in a manner such that it matches the measured solar readings. We multiply each data points of the luminance with a conversion factor, such that the distance between corresponding inter-samples of luminance and weather station is minimized (cf. Appendix A for details). We use this normalization factor, in order to map the computed image luminance to follow the similar cosine clear sky model. We observe that our computed luminance from the whole-sky image and the measured solar radiation closely follows each other. We emphasize here that it is an important contribution to successfully track the rapid solar fluctuations. Unlike other solar estimation models based on meteorological sensor data, our proposed model can successfully estimate the *peaks* and *troughs* of solar readings accurately.

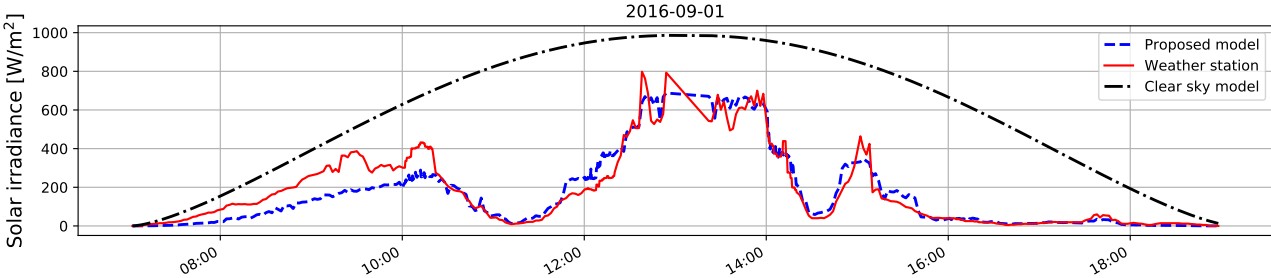

**Figure 8.** We show the measured weather station data (in red), and the clear sky radiation (in black) as on 01-Sep-2016. We observe that the image luminance normalized w.r.t. the measured solar irradiance, follows the measured readings closely. The sampling interval between two measurements is 2 minutes.

Using our proposed methodology, we compute the luminance of all the captured images. We use Eq. 3, and estimate the corresponding solar radiation values. The estimated solar irradiance value is compared with the actual irradiance values obtained from the solar sensors. The actual ones are recorded in the collocated weather station. These recordings serve as the ground-truth measurements. Figure 9 shows the histogram of difference between the estimated and actual solar radiation. We observe that the estimated solar radiation do not deviate much from the actual solar radiation. It is clear that $47.9\%$ of data points are concentrated in the range $[-100, +100]$ W/m$^2$.

## 4.2 Benchmarking techniques

We benchmark our proposed approach with other existing solar estimation models. To the best of our knowledge, currently, there are no proposed models to estimate short-term fluctuations of solar irradiance from ground-based images. However, most remote sensing analysts have been using other meteorological sensor data eg. daily temperature, humidity, rainfall and dew point temperature to estimate daily solar irradiance. One of the pioneer work was done by Hargreaves and Samani (Hargreaves and Samani, 1985), who proposed a model based on daily temperature variations. Donatelli and Campbell (Donatelli and Campbell, 1998) improved the model by including clear sky transitivity as one of the factors. On the other hand, Bristow and Campbell (Bristow and Campbell, 1984) also proposed a new model of solar radiation estimation, by including the atmospheric transmission coefficient. Subsequently, Hunt *et al.* (Hunt et al., 1998) showed that the solar estimation model can be further

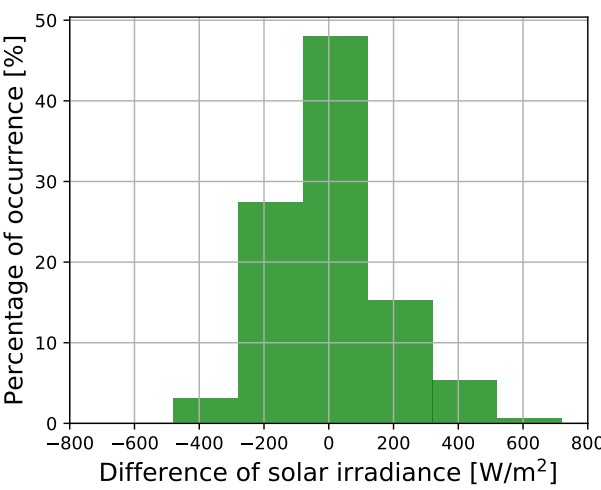

**Figure 9.** Histogram of difference between estimated and actual solar irradiance. We observe that most of the data are concentrated in the 0-bin.

improved by incorporating the daily precipitation data in the model. We benchmark our proposed approach with these different existing models. We illustrate the various benchmarking models in Fig. 10. Unfortunately, most of these algorithms fail to capture the short-term variations of the actual solar radiation.

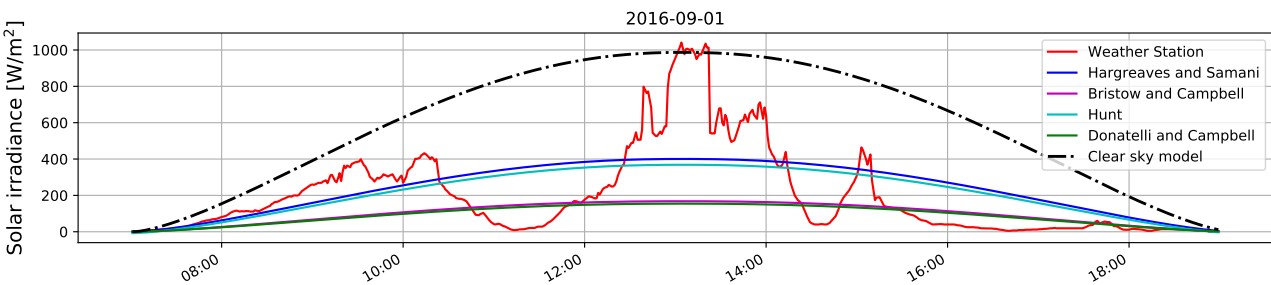

**Figure 10.** Comparison amongst different benchmarking solar estimation models, along with clear sky model and measured solar irradiance on 01-Sep-2016. We observe most of the existing algorithms fail to capture the rapid fluctuations of the measured solar irradiance.

We calculate the Root Mean Square Error (RMSE) of the estimated solar radiation and Spearman's rank correlation coefficient as the evaluation metrics. The RMSE of an estimation algorithm represents the standard deviation of the actual and estimated solar radiation values. Table 2 shows the RMSE values of our proposed algorithm with the other existing benchmarking algorithms. Our proposed approach performs the best. We also evaluate the spearman correlation coefficients of the different benchmarking algorithms, since this is a non-parametric measure to find the relationship between measured and estimated solar radiation. This does not assume that the underlying dataset are derived from a normal distribution. We report the

correlation values in Table 2. Our proposed approach has also the highest correlation amongst all methods. Table 2 explains the results where the training and testing set of images are identical, and all images are considered for evaluation.

| Methods | RMSE (W/m$^2$) | Correlation |
|---|---|---|
| Proposed approach | 178.27 | 0.86 |
| Hargreaves and Samani | 982.35 | 0.67 |
| Bristow and Campbell | 318.07 | 0.68 |
| Donatelli and Campbell | 324.48 | 0.67 |
| Hunt *et al.* | 922.66 | 0.65 |

**Table 2.** Benchmarking of our proposed approach with other solar radiation estimation models. All correlation values have p-value equal to 0.

Furthermore, we are also interested to check if our proposed model can generalize well with random samples of our captured sky camera images. We choose a random selection of images as the training set, and fit our linear regressor on these selected training images. The RMSE values are then calculated on these training images. We perform this analysis for varying percentage of training images. Each experiment is performed 100 times to remove any selection bias.

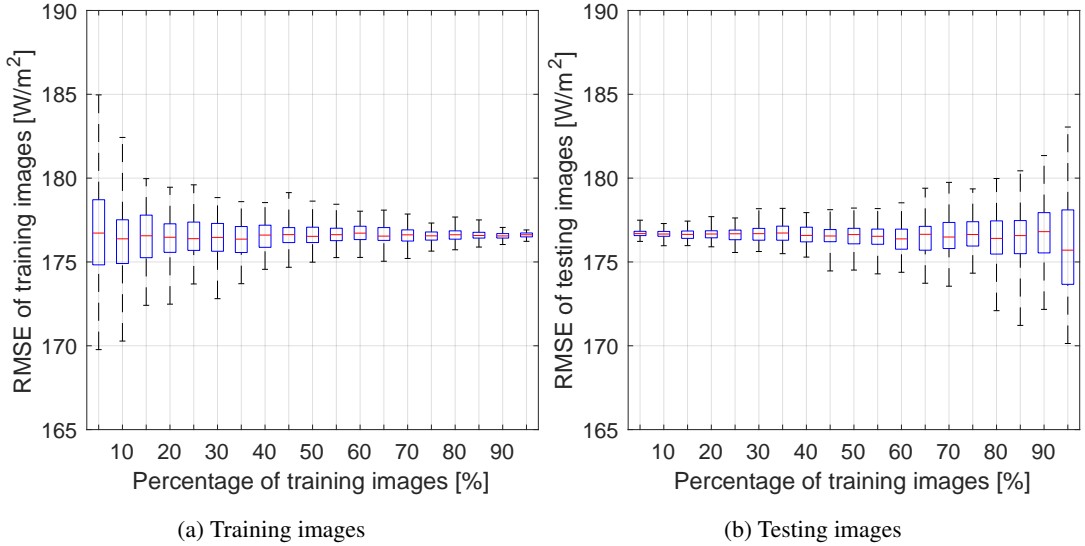

(a) Training images          (b) Testing images

**Figure 11.** Effect of the percentage of training images on RMSE values. The lower and upper end of each box represents the $25^{th}$ and $75^{th}$ percentile of the data, and the red line represents the median value. Each experiment is conducted 100 times with a random choice of training and testing sets.

Figure 11(a) shows the results on training images. We observe that the variation of the RMSE values gradually decreases, as we increase the number of training images. Moreover, we also check the variation of RMSE values when the testing images

are not identical as training images. Once we choose a random selection of images as training set, the remaining images are considered as the testing set. We show the RMSE results on such images in Fig 11(b). As expected, the variation of RMSE values increases with higher percentage of training images. The linear regressor model overfits the data, and provides higher variation in the error when tested on a fewer testing images. However, the average RMSE does not vary much in all cases. Therefore, our proposed model is free from selection bias, and generalizes well with random selection of training and testing images.

We represent the scatter plot between the measured solar radiation and estimated solar radiation for the different benchmarking algorithms in Fig. 12.

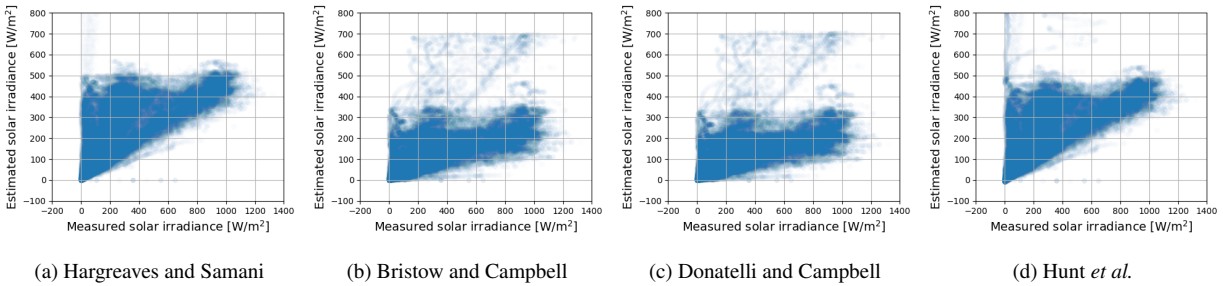

| (a) Hargreaves and Samani | (b) Bristow and Campbell | (c) Donatelli and Campbell | (d) Hunt *et al.* |

**Figure 12.** Scatter plot between measured solar irradiance and estimated solar irradiance for the benchmarking algorithms.

We observe that there is no strong correlation for most of these existing algorithms. This is because meteorological sensor data alone, with no cloud information cannot determine the sharp fluctuations of the solar radiation. This is an important limitation of these models, which we attempt to address in this paper. Our model based on sky images have additional information about cloud movement and its evolution, which is the fundamental reason behind rapid solar radiation fluctuations. In our proposed model, most of these short-term variations are captured properly (cf. Fig. 8).

## 5 Discussion

### 5.1 Short-term forecasts

Our proposed approach can estimate the solar radiation accurately with the least root mean square error, as compared to other models. The main advantage of our approach is that it can be used on predicted images as well, opening the potential for short term solar irradiance forecasting, which is needed in the solar energy field. As an initial case study, we have exploited optical flow techniques to estimate the direction and flow of cloud motion vectors between two successive image frames. We use the $(B - R)/(B + R)$ ratio channel of the sky/cloud image, where $B$ and $R$ are the blue and red channels respectively. We use an implementation of optical flow technique (flo) that uses a simpler conjugate gradient solver to obtain the flow field. Figure 13 illustrates the estimated flow field.

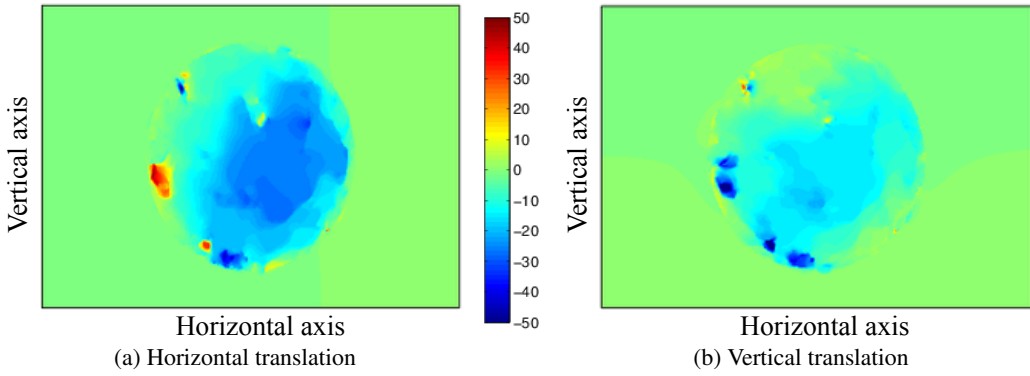

**Figure 13.** We visualize the horizontal and vertical translation of pixels between two successive frames, using optical flow technique.

Using the images captured at $t$ and $t-2$ minutes, we estimate the horizontal- and vertical- translation of the pixels. Under the assumption that the flow of cloud motion vectors for the successive $t+2$ minutes is similar to that of previous frames, we estimate the future $t+2$ minutes frame, and subsequently the $t+4$ minutes frame. Figure 14 illustrates this. We obtain a forecast accuracy of $70\%$ for a prediction lead time upto 6 minutes.

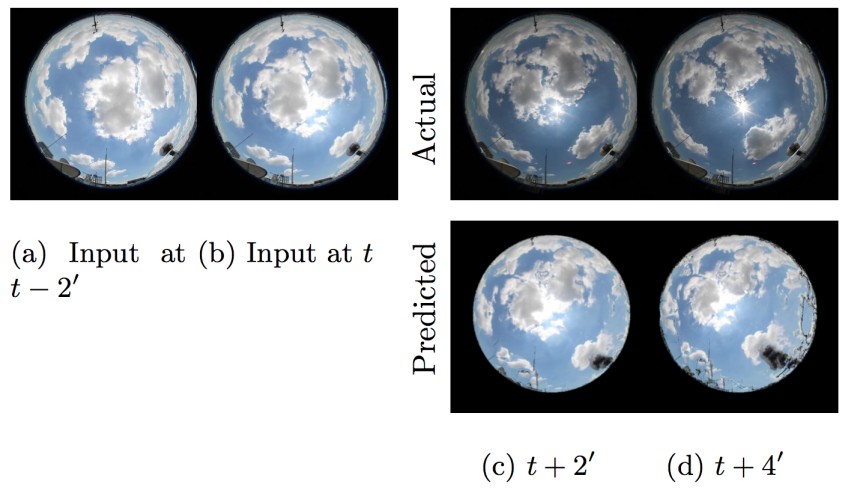

**Figure 14.** Prediction of sky/cloud image using optical flow technique.

5    In the future work, we will use our proposed methodology of estimating solar irradiance on this predicted sky/cloud image. Such approach will provide us more stable and reliable forecasts of solar irradiance.

## 5.2 Scope of improvement

This paper proposes an empirical model to estimate the solar irradiance from ground-based cameras. However, there are scopes of improvement in our approach. In this section, we highlight the issues and suggest techniques to address them. Firstly, we use *JPEG* images instead of uncompressed *RAW* images for the computation of scene luminance. The *JPEG* compression algorithm introduces non-linearities in the pixel values and our proposed model thus deviates from a linear relationship. We can generate more consistent results by using only *RAW* format images. Nevertheless, we still use *JPEG* images, as they have a significantly smaller size and less perceptible distortion in image quality. This assumption is practical from an operational point of view. On the other hand, uncompressed *RAW* images have large file size and it is impossible to capture and store *RAW* images at short capturing intervals due to the induced latency. In our future work, we intend to use *RAW* format images for the computation of solar irradiance values from sky cameras.

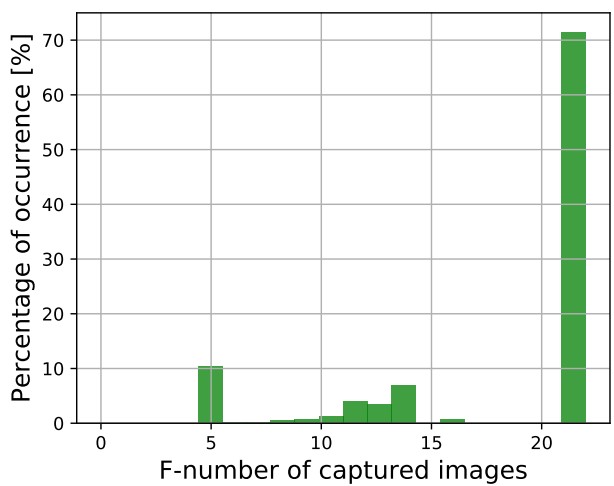

**Figure 15.** Distribution of F-number of the WAHRSIS images that are used to derive the proposed model.

Secondly, our captured images have a wide range of camera settings with varying shutter speed, ISO and aperture values. This is disadvantageous because the relationship between pixel value and camera aperture value becomes non-linear for larger F-numbers. The relationship deviates from linearity above F 4.0 (Hiscocks and Eng, 2011). Figure 15 depicts the wide range of F-numbers in the captured images used in deriving our proposed model. We observe that a significant percentage of images have large F-numbers, where the non-linearity sets in. This can be solved by using the aperture priority mode of the sky camera, wherein the F-number is fixed, and the exposure time varies dynamically to match the lighting conditions of the scene.

## 6    Conclusion & Future work

We presented a method for estimating the rapid fluctuations of the solar irradiance using the luminance of images taken by a whole sky imager. We are able to estimate the sharp short-term variations, which significantly improves the state-of-the-art. This approach is of interest in the solar energy field, because these variations cause a sudden decrease in the electricity generation from solar panels. Short-term predictions of such ramp-downs are needed to maintain the stability of the power grid. Combining our solar irradiance estimation approach with cloud movement tracking in the input images could ultimately lead to better irradiance predictions. Such information on rapid fluctuations of solar irradiance can assist in establishing a high-reliability solar energy generation system. We also plan to explore methodologies from time-series modelling (Dev et al., 2018a), to predict solar irradiance.

## 7    Code availability

The source code of all simulations in this paper is available here: https://github.com/Soumyabrata/estimate-solar-irradiance.

## Appendix A:  Derivation of normalization factor

Let us suppose that $a_1$, $a_2$, ..., $a_t$ be the weather station records for $t$ number of time stamps. The luminance values computed for each of the corresponding weather station points are represented by $b_1$, $b_2$, ..., $b_t$. We attempt to estimate the conversion factor $x$, such that the objective function $f(x)$ representing the inter-sample distances between weather station and computed luminance value is minimized.

We represent objective function $f(x)$ as:

$$f(x) = (xb_1 - a_1)^2 + (xb_2 - a_2)^2 + \ldots + (xb_t - a_t)^2$$
$$= \sum_{i=1}^{t} (xb_i - a_i)^2$$

We equate $f'(x)$ to 0, and the normalization factor $x$ is found as $x = \frac{\sum_{i=1}^{t} a_i b_i}{\sum_{i=1}^{t} b_i^2}$.

*Acknowledgements.* This research is funded by the Defence Science and Technology Agency (DSTA), Singapore. The ADAPT Centre for Digital Content Technology is funded under the SFI Research Centres Programme (Grant 13/RC/2106) and is co-funded under the European Regional Development Fund.

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
