# Peer review of "Estimating Solar Irradiance Using Sky Imagers"

_Atmospheric Measurement Techniques, 2019_

## Referee Comment (RC1) · Anonymous Referee #1 · 31 May 2019

**Review comments on "Estimating Solar Irradiance Using Sky Imagers" (Manuscript ID amt-2019-141)**

This paper proposes a new model for estimating the solar irradiance using pictures. This method has some new ideas. In particular, this method may play an important role in the prediction of solar radiation. But, It seems to me that this research shows a kind of preliminary results, so more details and data are required to extract robust conclusions. Therefore, this version of the manuscript is not ready for a regular article.

General comments.

1. The most important problem to be explained is the calibration. Because the parameters of each WSIs are different, the new WSIs must be calibrated for six months to one year to achieve solar radiation estimation. This is very limited in practical application. Even so, what is the uncertainty of estimation result?

2. The method of determining the sampling points around the sun proposed in this paper is not the core issue in my opinion. In fact, with the sun as the center, the result of determining the sampling point by any method based on distance weighting will not be very different from the result in this paper. Or, the results of these methods should be compared in the paper.

3. Based on the current research, the paper hopes to further realize the prediction of solar radiation based on pictures obtained by WSIs. That's really a good idea. However, the predicted results should also be given in the article. Because if there is no next step to predict radiation, this paper has no practical application value. (Solar pyranometers are cheaper and more accurate than WSIs)

4. The author should pay attention to the prediction of solar radiation, Especially in the first step, cloud motion prediction. There are many problems in cloud motion prediction based on distorted images. How accurate the radiation prediction can be obtained from the predicted image should be explained together.

**Specific comments**

5. The references cited in this paper are incorrect.

6. Some abbreviations need to be given in full English and even explained. For example, DSLR (P3L29);

7. P4L4: Davis Instruments 7440 Weather Vantage Pro, References or detailed explanations are required.

8. How did the P6L9 formula come into being? Is it suitable for use here? Explanation is needed

9. P12, Figure 7. "Watt/m2" should be "Watt/m$^2$".

---

## Referee Comment (RC2) · Anonymous Referee #2 · 21 Jun 2019

Summary:

This paper presents a method to quantify the solar irradiance by using an all-sky imager with a fish eye lens. This uses data sampled in Singapore, with variations in cloudy scenes, and is aimed at better prediction of photovoltaic energy production. This paper describes a model by which a sky image's individual pixels, from compressed jpg, are used to related to a The paper is mostly well written, and concise, bordering on too short. The approach is interesting, and the applications may be wide ranging for energy forecasting. That said, the model is not well described, with major issues concerning the non-linearity and the normalization factor, that seems to make the model results look better than actuality. Model here is also a bit of a stretch as it is simply an empirical fit of the measured values. The actual evaluation of the model requires a normalization

factor based on the difference to what is considered truth, the irradiance pyranometer measurements. There are some instances of colloquial use of English language, while most of the citations are not well formatted. The large use of footnotes also seems to be distracting to the content.

The benchmarking technique against other models seems to be lacking most of the state-of-the-art techniques that can easily be found through literature review. Techniques described by Chu et al., 2015, Baharin et al., 2016 for all sky imagers, or satellite-based methods such as the start from Mueller et al., 2004. There are likely many more methods, but just to name a few not reference in this manuscript.

Because of the questions on the model assumptions, and its evaluation against dated models, this manuscript does not seem to meet the criteria for publication unless there is significant rework. The simplistic approach does seem to have merit, but it may be overstated.

General Comments:

1. The abstract details 'accurate' but a root mean square deviation of 178 W/mˆ2 is hardly accurate. The other 'state of the art' methods should be described or at least mentioned.

2. Throughout the manuscript, the citations are not in the AMT style. They are often mostly added to end of sentences without proper brackets.

3. The use of random points in the modelling of the solar irradiance is not well described. Why not use the entire image, and subsetting by using a cloud mask?

4. The irradiance definition is not commonplace for atmospheric research: "The first step in estimating irradiance from the luminance is thus to cosine weight it according to its direction of flow" (p. 7 line 22-23). It should be cosine weighted per regards to the normal of a horizontal plane.

5. Footnotes should be either incorporated directly into the text, or as citations. Many

caveats and omissions are found in the footnotes that should be more explicitly presented.

6. Since much of the manuscript relies on comparing to pyranometer measurements, a more in-depth description of such instrument should be included.

7. The discussion of the non-linearity seems to be rather lacking of analytical analysis.

8. The normalization factor depends on having both irradiance and all sky imaging – this seems to be hiding the discrepancies of the model. Leaves the reader doubting the results from figure 4 and 5.

Specific Comments:

9. P.2 lines 3 – 13, missing multiple citations for MODIS products, GEOSS, Ouarda et al. citation is oddly written. SEVIRI acronym not defined.

10. P.2 line 23, What are "hot belts" ? This regional language use should be amended for a more understood English sentence.

11. P.6, footnote could be inserted into text, also there is no reference to "SMPTE recommended Partice 177 "

12. P. 7, footnote should be a full citation and inserted within text.

References:

Baharin, K. A., Abdul Rahman, H., Hassan, M. Y. and Gan, C. K.: Short-term forecasting of solar photovoltaic output power for tropical climate using ground-based measurement data, J. Renew. Sustain. Energy, 8(5), 53701, doi:10.1063/1.4962412, 2016.

Chu, Yinghao & Urquhart, Bryan & Gohari, S M Iman & Pedro, Hugo & Kleissl, Jan & F.M. Coimbra, Carlos: Short-term reforecasting of power output from a 48 MWe solar PV plant. Solar Energy. 112. 68-77. 10.1016/j.solener.2014.11.017, 2015.

Mueller, R. W., Dagestad, K. F., Ineichen, P., Schroedter-Homscheidt, M., Cros,

S., Dumortier, D., Kuhlemann, R., Olseth, J. A., Piernavieja, G., Reise, C., Wald, L. and Heinemann, D.: Rethinking satellite-based solar irradiance modelling: The SOLIS clear-sky module, Remote Sens. Environ., 91(2), 160–174, doi:https://doi.org/10.1016/j.rse.2004.02.009, 2004.

---

## Author Comment (AC1) · 10 Aug 2019

The comment was uploaded in the form of a supplement:
https://www.atmos-meas-tech-discuss.net/amt-2019-141/amt-2019-141-AC1-supplement.pdf
* * *

---

## Author Response (AR1)

**Response to Reviewers' Comments**
**amt-2019-141: Estimating Solar Irradiance Using Sky Imagers**

Soumyabrata Dev, Florian M. Savoy, Yee Hui Lee, Stefan Winkler

August 10, 2019

We would like to thank the Associate Editor and the anonymous reviewers for your valuable comments and suggestions. Based on your inputs, we have thoroughly revised the manuscript. All the comments and suggestions have been addressed and implemented in this revised manuscript.

Responses to the individual comments can be found below. Unless otherwise specified, the references, equations, figures and tables cited in the answers are numbered as per the revised manuscript.

```
>> REVIEWER 1<<
This paper proposes a new model for estimating the solar irradiance using pictures. This
    method has some new ideas. In particular, this method may play an important role in
    the prediction of solar radiation. But, It seems to me that this research shows a
    kind of preliminary results, so more details and data are required to extract robust
    conclusions. Therefore, this version of the manuscript is not ready for a regular
    article.
```

Thanks for the comment. The core idea of this manuscript is to establish the fact that images obtained from ground-based sky cameras can assist us in accurately estimating the rapid fluctuations of measured solar irradiance. This is the first step, for a robust and reliable short-term solar forecasting. In this manuscript, we restrict our discussions to mainly solar irradiance estimation, and do not delve into solar irradiance prediction.

We have incorporated several major changes in the current version of the manuscript. Some of the major changes include:

- We have provided a detailed discussion of the calibration techniques – white balancing, geometric calibration and vignetting correction, used in our ground-based sky camera;

- We have provided an extensive evaluation of several other non-linear models (or empirical fits) between solar irradiance and image luminance. Instead of a linear model, we have investigated the use of a higher-order polynomial fit to model the irradiance based on image-based luminance. More discussions of these models are provided in Section 3.3 of the revised manuscript; and

- We have also added useful insights on the use of optical flow techniques to forecast the future sky/cloud image. We provided our initial results on estimating the image forecasts with a lead time of upto 15 minutes. We discussed this in Section 5 of the revised manuscript.

In the revised manuscript, these changes are incorporated throughout the manuscript.

```
General comments.
```

1.The most important problem to be explained is the calibration. Because the parameters
   of each WSIs are different, the new WSIs must be calibrated for six months to one
   year to achieve solar radiation estimation. This is very limited in practical
   application. Even so, what is the uncertainty of estimation result?

Thanks for the comment. We agree that the calibration of the ground-based cameras is an important step. Our ground-based camera WAHRSIS is calibrated with respect to – white balancing, geometric calibration and vignetting correction.

The imaging system in WAHRSIS is modified so that it captures the near-infrared region of the spectrum. Hence, the red channel of the captured image is more prone to saturation. It renders the captured image reddish in nature. Therefore, we employ custom white balancing in the camera, such that it compensates the alteration owing to the near-infrared capture. Figure 1 depicts the captured images obtained from automatic and custom white balancing.

[Figure]

(a) With auto white-balancing      (b) With custom white-balancing

Figure 1: We use custom white-balancing for correcting the white balance.

We use the popular toolbox by Scaramuzza *et al.* [1] for the geometric calibration of WAHRSIS. This process involves the computation of the intrinsic parameters of the camera. We use a black-and-white regular checkerboard pattern, and position it at various positions around the sky camera. Figure 2 illustrates a few sample positions of the checkerboard. Using user interaction to identify the corner points and the known 3D co-ordinates, we estimate the intrinsic parameters of the camera.

[Figure]

Figure 2: We position the checkerboard at various locations for the geometric calibration.

Finally, we also employ vignetting correction to the images captured by our sky camera. Owing to the fish-eye nature of the lens, the area around the centre of the lens is brighter, as compared to the sides. We use an integrating sphere to correct this variation of illumination. Figure 3 depicts an image captured inside an integrating sphere that provides an uniform illumination distribution in all directions. We use this reference image to correct the illumination of all captured sky/cloud images by our sky camera.

[Figure]

Figure 3: We captured a reference image inside the uniformly-illuminated integrating sphere.

Of course, these calibration techniques are fundamental, and needs to be completed for subsequent analytics using sky cameras. In the revised manuscript, we have mentioned that a prior calibration of the imaging systems is required for estimating solar irradiance from sky cameras.

In the revised manuscript, we have added this discussion in Section 2.1 of the manuscript.

2. The method of determining the sampling points around the sun proposed in this paper is not the core issue in my opinion. In fact, with the sun as the center, the result of determining the sampling point by any method based on distance weighting will not be very different from the result in this paper. Or, the results of these methods should be compared in the paper.

Thanks for the comment in investigating an alternate strategy of sampling. In our previous work [2], we used a cropped version of the image with the sun as the centre. However, we realised that there are several disadvantages to this approach.

- We need to find the position of the sun in the image, in order to crop a square image with the sun as centre. However, this identification becomes difficult during overcast, wherein the sun is completely covered by clouds;

- We also need to ascertain the optimal crop size dimension to obtain the best accuracy. We compute the correlation value between solar irradiance value and image luminance value for various crop size dimensions. Figure 4 shows the impact of crop size on the obtained correlation.

[Figure]

Figure 4: We observe that the best performance is obtained for a crop size of $300 \times 300$ (red bar).

Therefore, in order to avoid these demerits, we sample the pixels around the hemispherical dome to estimate the solar irradiance values. Our obtained results (cf. Section 4) establish that such sampling strategy work well in estimating solar irradiance.

In the revised manuscript, we have added this discussion in Section 3.1 of the manuscript.

```
3.Based on the current research, the paper hopes to further realize the prediction of
   solar radiation based on pictures obtained by WSIs. That's really a good idea.
   However, the predicted results should also be given in the article. Because if there
   is no next step to predict radiation, this paper has no practical application value.
   (Solar pyranometers are cheaper and more accurate than WSIs)
```

Thanks for the feedback. We absolutely agree that the full potential of using ground-based sky cameras will be realised in the solar energy *prediction* stage, in addition to the current solar irradiance *estimation*. The main focus of this manuscript is to establish to our community that ground-based imaging systems can provide us useful insights in understanding and estimating the rapid fluctuations of solar irradiance over the different hours of the day. In this paper, we restricted our message to convey this key idea of solar irradiance estimation, and did not delve into solar irradiance forecasting.

However, in this response letter, we provide our initial result on solar irradiance forecasting. We borrow techniques from time series modelling, to forecast future solar irradiance values. In our recent work [3], we modelled the solar irradiance values using triple exponential smoothing (TES). Figure 5 illustrates the performance of TES model for a short lead time. These solar irradiance recordings are measured in the interval of 1 minute. We use a historical data of 1000 observations, to estimate the next 50 observations. We set the seasonal period in TES model as 288. We observe that the predicted solar irradiance values (represented in green color) closely follows the actual solar irradiance values.

[Figure]

Figure 5: We illustrate the clearness index (k) values in the time series. This clearness index is defined as the ratio of observed solar irradiance and measured solar irradiance. We observe that the predicted solar irradiance values closely matches the actual solar irradiance.

In our upcoming work, we are currently incorporating the TES modelling technique directly on the sky camera image, instead of applying onto solar irradiance point measurements. We hope this will provide a better solar estimation technique than the current state-of-the-art methodologies.

In the revised manuscript, we have added this discussion in Section 6 of the manuscript.

4. The author should pay attention to the prediction of solar radiation, Especially in the first step, cloud motion prediction. There are many problems in cloud motion prediction based on distorted images. How accurate the radiation prediction can be obtained from the predicted image should be explained together.

Thanks for the feedback. We agree that the prediction of cloud motion vectors [4] is the first step in the realm of solar radiation forecasting. We have exploited optical flow techniques to estimate the direction and flow of cloud motion vectors between two successive image frames. We use the $(B - R)/(B + R)$ ratio channel of the sky/cloud image, where $B$ and $R$ are the blue and red channels respectively. We use an implementation [1] of optical flow technique that uses a simpler conjugate gradient solver to obtain the flow field. Figure 6 illustrates the estimated flow field.

[Figure]

(a) Horizontal translation        (b) Vertical translation

Figure 6: We visualize the horizontal and vertical translation of pixels between two successive frames, using optical flow technique.

Using the images captured at $t$ and $t - 2$ minutes, we estimate the horizontal and vertical translation of the pixels. Under the assumption that the flow of cloud motion vectors for the successive $t + 2$ minutes is similar to that of previous frames, we estimate the future $t + 2$ minutes frame, and subsequently the $t + 4$ minutes frame. Figure 7 illustrates this. We obtain a forecast
* * *
[1] Available at `https://people.csail.mit.edu/celiu/OpticalFlow/`

accuracy of 70% for a prediction lead time upto 6 minutes. However, the accuracy decreases for a longer lead time. In our future work, we will estimate the solar irradiance value on the predicted sky/cloud image, using the empirical model proposed in this manuscript.

[Figure]

(a) Input at $t - 2'$  (b) Input at $t$

(c) $t + 2'$   (d) $t + 4'$

Figure 7: Prediction of sky/cloud image using optical flow technique.

In the revised manuscript, this discussion is added to Section 5.1 of the manuscript.

```
Specific comments
5.The references cited in this paper are incorrect.
```

Thank you for pointing out. In the revised manuscript, we have ensured that all the references are cited in the AMT style.

In the revised manuscript, these changes are incorporated throughout the manuscript.

```
6.Some abbreviations need to be given in full English and even explained. For example,
    DSLR (P3L29);
```

Thanks for the comment. In the revised manuscript, we have provided the full form of digital single-lens reflex (DSLR). We have also provided detailed explanations of DSLR camera.

In the revised manuscript, we have provided the changes in Section 2.1 of the manuscript.

```
7.P4L4: Davis Instruments 7440 Weather Vantage Pro, References or detailed explanations
    are required.
```

Thanks for the feedback. We have now added more discussion in the revised manuscript. The solar pyranometer is included in *Davis Instruments 7440 Weather Vantage Pro* weather station. It measures the total solar irradiance flux density in Watt/m$^2$. On a clear day with no occluding clouds, the solar sensor ideally follows a typical cosine response. Figure 8 shows the theoretical response of the solar sensor in the pyranometer, for varying degrees of solar incident angle.

[Figure]

Figure 8: Response of the solar sensor with varying solar incident angle.

In the revised manuscript, the changes are made in Section 2.2 of the manuscript.

8.How did the P6L9 formula come into being? Is it suitable for use here? Explanation is
    needed

Thanks for the comment. The P6L9 formula is used to compute the luminance of an image from the $R$, $G$ and $B$ values of the $RGB$ image. The formula is proposed in SMPTE Recommended Practice 177 [5].

In the revised manuscript, we added this discussion in Section 3.2 of the manuscript.

9.P12, Figure 7. Watt/ m2   should be  Watt / m2

Thanks for pointing it out. The superscript in the y-label of the corresponding figure is now depicted properly.

In the revised manuscript, this update is performed in Section 4.2 of the manuscript.

Soumyabrata Dev, Florian M. Savoy, Yee Hui Lee, Stefan Winkler

August 10, 2019

We would like to thank the Associate Editor and the anonymous reviewers for your valuable comments and suggestions. Based on your inputs, we have thoroughly revised the manuscript. All the comments and suggestions have been addressed and implemented in this revised manuscript.

Responses to the individual comments can be found below. Unless otherwise specified, the references, equations, figures and tables cited in the answers are numbered as per the revised manuscript.

```
>> REVIEWER 2<<
This paper presents a method to quantify the solar irradiance by using an all-sky imager
    with a fish eye lens. This uses data sampled in Singapore, with variations in cloudy
    scenes, and is aimed at better prediction of photovoltaic energy production. This
    paper describes a model by which a sky images individual pixels, from compressed jpg,
    are used to related to a
```

Thanks for your positive feedback to this manuscript. We believe that an accurate estimation of the solar irradiance using the ground-based cameras is the first step for a reliable short-term solar forecasting.

```
The paper is mostly well written, and concise, bordering on too short.
```

Thanks for your positive feedback on the writing style of the manuscript. In this manuscript, we kept our message clear and concise on the use of sky cameras for solar irradiance estimation.

In the revised manuscript, we have added more extensive details and explanations. Some of the major changes include:

- We have provided a detailed discussion of the calibration techniques – white balancing, geometric calibration and vignetting correction, used in our ground-based sky camera;

- We have provided an extensive evaluation of several other non-linear models (or empirical fits) between solar irradiance and image luminance. Instead of a linear model, we have investigated the use of a higher-order polynomial fit to model the irradiance based on image-based luminance. More discussions of these models are provided in Section 3.3 of the revised manuscript; and

- We have also added useful insights on the use of optical flow techniques to forecast the future sky/cloud image. We provided our initial results on estimating the image forecasts with a lead time of upto 15 minutes. We discussed this in Section 5 of the revised manuscript.

In the revised manuscript, these changes are incorporated throughout the manuscript.

```
The approach is interesting, and the applications may be wide ranging for energy
    forecasting. That said, the model is not well described, with major issues concerning
    the non-linearity and the normalization factor, that seems to make the model results
    look better than actuality. Model here is also a bit of a stretch as it is simply an
    empirical fit of the measured values.
```

Thanks for the comment.

The main idea of this manuscript is to establish that images obtained from sky camera can be used to estimate the rapid fluctuations of solar irradiance observed throughout the day. The theoretical clear sky model follows a cosine response. In order to map the computed image luminance to follow the similar cosine clear sky model, we use the normalization factor.

In the revised manuscript, we have provided more details about our proposed methodology. As suggested by you, we renamed *model* to *empirical fit*, in order to better convey our proposed message. In the revised manuscript, we provided other non-linear empirical fits between image luminance and solar irradiance. Figure 1 describes the various linear and non-linear fits, using our proposed methodology. We observe that the higher-order polynomials (quartic and quintic) are highly ill-conditioned.

[Figure]

(a) Linear fit          (b) Quadratic fit

(c) Cubic fit

(d) Quartic fit          (e) Quintic fit

Figure 1: Empirical fit between solar radiation and image luminance computed with our proposed framework. We observe that it deviates from linearity at higher luminance values. Also, the higher order polynomials are ill-conditioned.

We also compute the Root Mean Square Error (RMSE) value between the actual- and regressed-values. Table 1 summarises the performance of the different order polynomials.

| Proposed models | RMSE (Watt/m$^2$) |
|---|---|
| Linear model (degree 1) | 178.27 |
| Quadratic model (degree 2) | 178.26 |
| Cubic model (degree 3) | 176.57 |
| Quartic model (degree 4) | 176.52 |
| Quintic model (degree 5) | 176.49 |

Table 1: Performance evaluation of various polynomial order regressors. We measure the RMSE value for each of the models.

We observe that lower order polynomials of degree 1 and 2 perform slightly inferior to those of higher order polynomials. We observe that the performance of cubic, quartic and quintic models are similar. We observe that the quartic and quintic models are highly ill-conditioned from Fig. 1. The cubic model is the best fit, therefore we propose this in our revised manuscript.

In the revised manuscript, these changes are incorporated in Section 3.3 of the manuscript.

```
The actual evaluation of the model requires a normalization factor based on the
    difference to what is considered truth, the irradiance pyranometer measurements.
```

Thanks for the feedback. During the evaluation stage, the estimated solar irradiance value is, in fact, compared with the actual irradiance values obtained from the solar sensors. This may not be very clear in the earlier version of the paper. We have re-emphasized the same in the revised manuscript.

In the revised manuscript, we have added this discussion in Section 4.1 of the manuscript.

```
There are some instances of colloquial use of English language, while most of the
    citations are not well formatted. The large use of footnotes also seems to be
    distracting to the content.
```

Thanks for this important feedback. We have revised the sentences in the revised manuscript, and also ensured that all citations are formatted in the AMT style. We have also reduced the number of usage of the citations throughout the paper.

In the revised manuscript, these changes are incorporated throughout the paper.

```
The benchmarking technique against other models seems to be lacking most of the state-of-
    the-art techniques that can easily be found through literature review. Techniques
    described by Chu et al., 2015, Baharin et al., 2016 for all sky imagers, or satellite
    -based methods such as the start from Mueller et al., 2004. There are likely many
    more methods, but just to name a few not reference in this manuscript. Because of the
     questions on the model assumptions, and its evaluation against dated models, this
    manuscript does not seem to meet the criteria for publication unless there is
    significant rework.
```

Thanks for your suggestions on the related work. We have added discussion of these publications in Section 1.1 of the revised manuscript. Baharin *et al.* proposed a machine-learning forecast model for PV power output, using Malaysia as the case study. Similarly Chu *et al.* used a reforecasting method to improve the PV power output forecasts with a lead time of 5, 10, and 15 minutes. Mueller *et al.* proposed a clear sky model that is based on radiative transfer models obtained from Meteosat's atmospheric parameters.

The focus of our manuscript is to estimate the solar irradiance values, from ground-based observations. We do not discuss about solar forecasts in this paper. Therefore, we have benchmarked with only those methods that attempt to estimate solar irradiance from ground-based observations. Both Baharin *et al.* and Chu *et al.* discuss solar forecast methodologies, and Mueller *et al.* discuss about satellite-based clear sky modelling.

```
The simplistic approach does seem to have merit, but it may be over stated.
```

Thanks for the comment. We indeed kept a simple approach to map the solar irradiance to the image-based luminance. Our idea is to establish that that ground-based sky cameras can provide

us more information about the evolution of clouds, as compared to point-based solar pyranometers. The luminance computed from sky camera images can match the rapid fluctuations of the measured solar irradiance.

General Comments:

1.  The abstract details 'accurate' but a root mean square deviation of 178 W/m^2 is hardly accurate. The other state of the art methods should be described or at least mentioned.

Thanks for the feedback. As suggested, we have edited the abstract of the revised manuscript in indicating the performance of the proposed methodology. We have also added the benchmarking methods in the abstract.

In the revised manuscript, these changes are indicated in the abstract of the manuscript.

2.  Throughout the manuscript, the citations are not in the AMT style. They are often mostly added to end of sentences without proper brackets.

Thanks for pointing it out. In the revised manuscript, we have ensured that citations are referenced in the AMT style.

In the revised manuscript, these changes are incorporated in all sections of the paper.

3.  The use of random points in the modelling of the solar irradiance is not well described. Why not use the entire image, and subsetting by using a cloud mask?

Thanks for the feedback. In our previous work [1], we used a cloud mask around the sun to estimate the solar irradiance. However, there are several disadvantages to the cloud mask approach:

- We need to find the position of the sun in the image, in order to crop a square image with the sun as centre. However, this identification of the sun in the image becomes difficult during overcast condition, wherein the sun is completely covered by clouds;

- We also need to ascertain the optimal crop size dimension to obtain the best accuracy. We compute the correlation value between solar irradiance value and image luminance value for various crop size dimensions. Figure 2 shows the impact of crop size on the obtained correlation.

[Figure]

Figure 2: We observe that the best performance is obtained for a crop size of $300 \times 300$ (red bar).

Therefore, in order to avoid these demerits, we sample the pixels around the hemispherical dome to estimate the solar irradiance values. Our obtained results (cf. Section 4) establish that such sampling strategy work well in estimating solar irradiance.

In the revised manuscript, we have added this discussion of alternative strategies in Section 3.1 of the manuscript.

4.  The irradiance definition is not commonplace for atmospheric research: "The first step
    in estimating irradiance from the luminance is thus to cosine weight it according to
    its direction of flow" (p. 7 line 22-23). It should be cosine weighted per regards
    to the normal of a horizontal plane.

Thanks for the comment. Based on empirical evidences, we use a cosine weight according to the direction of flow. This provided us the best results as compared to other strategies, and hence we proceeded with the same.

In the revised manuscript, we have added this discussion in Section 3.3 of the manuscript.

5.  Footnotes should be either incorporated directly into the text, or as citations. Many
    caveats and omissions are found in the footnotes that should be more explicitly
    presented.

Thanks for the feedback. In the revised manuscript, we have ensured to use the minimum number of footnotes. Most of the citations are removed, and included directly onto the text.

In the revised manuscript, these changes are incorporated throughout the manuscript.

6.  Since much of the manuscript relies on comparing to pyranometer measurements, a more
    in-depth description of such instrument should be included.

Thanks for the feedback. We have now added more discussion of the solar pyranometer in the revised manuscript. The solar pyranometer is included in *Davis Instruments 7440 Weather Vantage Pro* weather station. It measures the total solar irradiance flux density in Watt/m$^2$. On a clear day with no occluding clouds, the solar sensor ideally follows a typical cosine response. Figure 3 shows the theoretical response of the solar sensor in the pyranometer, for varying degrees of solar incident angle.

[Figure]

Figure 3: Response of the solar sensor with varying solar incident angle.

In the revised manuscript, the changes are made in Section 2.2 of the manuscript.

7. The discussion of the non-linearity seems to be rather lacking of analytical analysis.

Thanks for the comment. We use *JPEG* images instead of uncompressed *RAW* images for the computation of scene luminance. The *JPEG* compression algorithm introduces non-linearities in the pixel values and, therefore our proposed empirical fit deviates from a linear relationship. We

can generate more consistent results by using only $RAW$ format images. In our future work, we intend to use $RAW$ format images for the computation of solar irradiance values from sky cameras.

In the revised manuscript, we have included this discussion in Section 5 of the manuscript.

8.  The normalization factor depends on having both irradiance and all sky imaging - this
    seems to be hiding the discrepancies of the model. Leaves the reader doubting the
    results from figure 4 and 5.

Thanks for the comment on the normalization factor. The theoretical clear sky model follows a cosine response. In order to map the computed image luminance to follow the similar cosine clear sky model, we use the normalization factor. The main idea in this manuscript is to establish an empirical model of the image luminance, that can estimate the rapid fluctuations of the solar irradiance.

In the revised manuscript, we have added this discussion in Section 4.1 of the manuscript.

Specific Comments:
9. P.2 lines 3  13, missing multiple citations for MODIS products, GEOSS, Ouarda et al.
   citation is oddly written. SEVIRI acronym not defined.

Thanks for the feedback. We have added references for MODIS [2] and GEOSS [3]. The Ouarda *et al.* citation is now fixed. We have also spelled out SEVIRI as Spinning Enhanced Visible and Infrared Imager.

In the revised manuscript, the changes are incorporated in Section 1 of the manuscript.

10. P.2 line 23, What are "hot belts" ? This regional language use should be amended for
    a more understood English sentence.

Thanks for the feedback. Our intended meaning is that tropical countries are more conducive for solar energy generation, because of the large amount of incident sunlight throughout the year. We have removed the phrase 'hot belts', and rephrased the sentence for no confusion.

In the revised manuscript, the changes are incorporated in Section 1.1 of the manuscript.

11. P.6, footnote could be inserted into text, also there is no reference to "SMPTE
    recommended Partice 177"

Thanks for the feedback. We have removed the corresponding footnote, and included it inline with the text. We have also added reference to the SMPTE Recommended Practice 177 [4] to the revised manuscript.

In the revised manuscript, the changes are incorporated in Section 3.2 of the manuscript.

12. P. 7, footnote should be a full citation and inserted within text.

Thanks for the feedback. The footnote in pp. 7 is now removed, and inserted within the text. In the revised manuscript, we have ensured to use the minimum number of footnotes. They are now mostly inserted within the texts, or omitted.

In the revised manuscript, the changes are incorporated in 3.3 of the revised manuscript.

[revised manuscript text omitted]